



# Evaluation of ozone trends in the mesosphere/lower thermosphere using a new merged dataset of ozone profiles

Monika E. Szelag[1], Viktoria F. Sofieva[1], Edward Malina[2], Pekka T. Verronen[1,3], Michelle L. Santee[4], Manuel López-Puertas[5], Bernd Funke[5], Gabriele Stiller[6], Alexandra Laeng[6], Kaley A. Walker[7], Patrick E. Sheese[7], Mark E. Hervig[8], Benjamin T. Marshall[9]

[1] Finnish Meteorological Institute, Helsinki, Finland
[2] ESA/ESRIN, Frascati, Italy
[3] Sodankylä Geophysical Observatory, University of Oulu, Sodankylä, Finland
[4] Jet Propulsion Laboratory, California Institute of Technology, Pasadena, California, USA
[5] Instituto de Astrofísica de Andalucía, CSIC, Granada, Spain
[6] Karlsruhe Institute of Technology, Karlsruhe, Germany
[7] Department of Physics, University of Toronto, Toronto, Canada
[8] GATS, Driggs, Idaho, USA
[9] GATS, Hampton, Virginia, USA

*Correspondence to*: monika.szelag@fmi.fi

**Abstract**. In recent years, the need for high-quality long-term mesospheric ozone records has become increasingly evident, as they are essential for understanding chemical, dynamical, and radiative processes in the middle and upper atmosphere and their coupling with the lower layers. Here, we present a new merged dataset of ozone profiles in the mesosphere and lower thermosphere (METEOR-$O_3$), created from several limb-viewing satellite instruments: HALOE, GOMOS, MIPAS, ACE-FTS, MLS, and SOFIE. The dataset covers the period from 1991 to 2023 and provides deseasonalized ozone anomalies in 10° latitude bins between 80°S and 80°N, from approximately 22 km to 100 km. The deseasonalized ozone anomalies are used for global and seasonal trend analysis. The results show positive upper stratospheric ozone trends in both hemispheres, with magnitudes of 1–2% per decade between 35 and 45 km, indicating continued ozone recovery consistent with previous assessments. In contrast, mesospheric ozone (above ~60 km) exhibits negative trends of 1–3% per decade, with the strongest decreases of about 8–10% per decade near the mesopause. Seasonal analyses confirm positive trends in the upper stratosphere across all seasons and persistent negative trends in the upper mesosphere, strongest at high latitudes above 80 km. The METEOR-$O_3$ dataset provides the first global, long-term merged record suitable for detailed studies of mesospheric/lower thermospheric ozone variability and trend evaluation, providing valuable information for model validation and assessments of upper atmospheric changes.



## 1    Introduction

Understanding long-term trends in the mesosphere and lower thermosphere (MLT) is becoming increasingly important as they can serve as a valuable indicator of climatic change. Trends in the MLT region exhibit significant variability which can be influenced by solar activity, atmospheric dynamics, and anthropogenic factors. Over the last years, the upper atmosphere has been widely studied, particularly concerning long-term trends. While temperature remains the most extensively studied, other parameters (winds, water vapour, minor constituents) have been analysed as well (Laštovička, 2023, Cnossen et al., 2024).

The dominant signal observed across datasets and models is cooling, primarily caused by the increasing concentration of $CO_2$ (Qian et al., 2021; Cnossen, 2020). Increases in $CH_4$ and $H_2O$, along with decreasing stratospheric O, contribute to cooling in the stratosphere and mesosphere, but their influence becomes negligible in the upper thermosphere (Cnossen et al., 2024 and references therein). The cooling trend has been observed in ground-based and satellite measurements with a magnitude of about 0.5-2 K per decade (Yuan et al., 2019; Bailey et al., 2021; Das, 2021; Li et al., 2021). Numerical simulations generally confirm the cooling trend except for the summer upper mesosphere, which shows near-zero or slightly positive temperature trends, likely reflecting dynamical effects (Qian et al., 2019; Solomon et al., 2019).

The long-term trends in dynamical processes, which are important in the MLT, are more complex. Observations and simulations reveal large spatial and temporal variability in wind trends, often differing between regions and seasons (Qian et al., 2019; Wilhelm et al., 2019). Research on atmospheric waves has been limited and concentrating mainly on tides (Laštovička, 2023). No significant long-term changes in diurnal tides have been observed, while semidiurnal components show mixed results that vary with altitude and latitude (Wilhelm et al., 2019). Overall, dynamical trends in the MLT are poorly constrained and regionally dependent, remaining a major source of uncertainty in understanding the MLT's long-term evolution (Laštovička, 2023 and references therein).

Mesospheric ozone plays an important role in atmospheric chemistry, dynamics, and energy balance, yet it remains one of the least-investigated parameters. Variations in its concentration can substantially influence the composition and behavior of the upper and middle atmosphere, thereby affecting the coupling between different atmospheric layers. In recent years, there has been a growing scientific interest in understanding solar-driven changes in ozone and temperature in the mesosphere and upper stratosphere. These variations, although occurring alongside the rapid climate change driven by anthropogenic greenhouse gas emissions, have the potential to regulate regional climate on annual to decadal timescales (Baumgaertner et al., 2011; Langematz et al., 2005; Maliniemi et al., 2014; Rozanov et al., 2005, 2012; Seppälä et al., 2009, 2013). Understanding these complex interactions between human-induced and naturally driven climate variability is particularly important in the polar regions, which are experiencing some of the most significant changes.

While individual satellite instruments provide ozone measurements in the mesosphere, their operation is limited in time. To date, a comprehensive merged data set combining multiple sources of mesospheric ozone data is not available, and an assessment of global mesospheric ozone trends has not been conducted. Previous analyses (Bizuneh et al., 2022; Huang et al.,





2014; Nath and Sridharan, 2014) based on 10-15 years of SABER (Sounding of the Atmosphere using Broadband Emission Radiometry) data were confined to narrow latitude ranges (5°N-15°N) or limited to lower and middle latitudes (48°S-48°N), thereby restricting insight into global long-term mesospheric ozone variability. Moreover, SABER ozone data are known to suffer from quality limitations (Mlynczak, private communication; López-Puertas et al., 2023), underscoring the need for improved observational datasets and longer time series to establish reliable long-term trends.

In this paper, we introduce a new merged dataset of ozone profiles in the mesosphere and the lower thermosphere, created using the data from several limb and occultation instruments: HALOE (Halogen Occultation Experiment), GOMOS (Global Ozone Monitoring by Occultation of Stars), MIPAS (Michelson Interferometer for Passive Atmospheric Sounding), ACE-FTS (Atmospheric Chemistry Experiment - Fourier Transform Spectrometer), MLS (Microwave Limb Sounder), and SOFIE (Solar Occultation For Ice Experiment). The paper is organized as follows. In Section 2, we provide short descriptions of the data used for the merged dataset. In Section 3, we describe the data preparation for the merging procedure, which is discussed in Section 4. Section 5 describes the new merged METEOR-O3 dataset. Section 6 is dedicated to trends analyses in the upper atmosphere. The summary (Section 7) concludes the paper.

## 2    Data

For the merged dataset, we used the limb and occultation instruments that provide data in the mesosphere and the lower thermosphere. Information about individual datasets is collected in Table 1. All datasets used to create the merged dataset have a vertical resolution of approximately 3-7 km in the MLT. Ozone measurements from the various satellite instruments provide global coverage and were obtained at different local times. The time series of the number of available ozone profiles per month from each instrument is shown in Figure 1. Note that for some instruments, the selected period is shorter than their full operational period; for example, MIPAS data are included starting from 2005, as in Sofieva et al. (2017, 2023). For instruments measuring in different illumination conditions (for example, daytime and nighttime measurements by MLS and MIPAS), the number of profiles per month is nearly identical. Latitudinal coverage of some of the datasets is illustrated in the Supplement (Figures S1 and S2). The best spatial coverage is provided by instruments with dense sampling (MLS and MIPAS), while solar occultation instruments contribute 14-15 sunrise and sunset measurements per day. SOFIE data are limited to the polar regions.

For GOMOS, MIPAS, and ACE-FTS, the data from the user-friendly HARMonized dataset of OZone profiles (HARMOZ) (Sofieva et al., 2013) are used. HARMOZ consists of the original retrieved ozone profiles from each instrument, screened for invalid data by the instrument experts and presented on a common vertical grid and in a common netCDF4 format, which simplifies the data usage. Below are more detailed descriptions of the individual datasets.

**Table 1: Information about the datasets used in the merged dataset.**



| Instrument/ satellite/ processor | Operation period | Local time/equator crossing time | Vertical range/ retrieval coordinate | Vertical resolution | Latitude coverage |
|---|---|---|---|---|---|
| **HALOE/UARS/v19** | Sep1991 – Nov 2005 | Sunset, sunrise | 250- 0.01 hPa / pressure | ~2 km | global |
| **GOMOS/ENVISAT/ ALGOM 2s v1** | Aug 2002 - Dec 2011 | 10 p.m. | 10-105 km / altitude | 3km | global |
| **MIPAS/Envisat KIT/IAA v8** | Jan 2002 - Apr 2012 | 10 a.m., 10 p.m. | MA UA: 20(40) -102 km NOM: 7-70 km / altitude | ~3-5km | global |
| **ACE-FTS/SCISAT/ v5.2** | Feb 2004 - present | Sunset, sunrise | 6-94 km / altitude | ~3-4km | global, best at mid-latitudes |
| **MLS/Aura/ NASA v5** | Aug 2004 - present | 1:45 a.m., 1:45 p.m. | 261-0.001 hPa (~9-96 km)/ pressure | ~3-7km | global |
| **SOFIE/AIM/ v1.3** | Apr 2007 - Mar 2023 | Sunset, sunrise | 50-100 km/altitude | ~2 km | high latitudes (65-85°) |

95

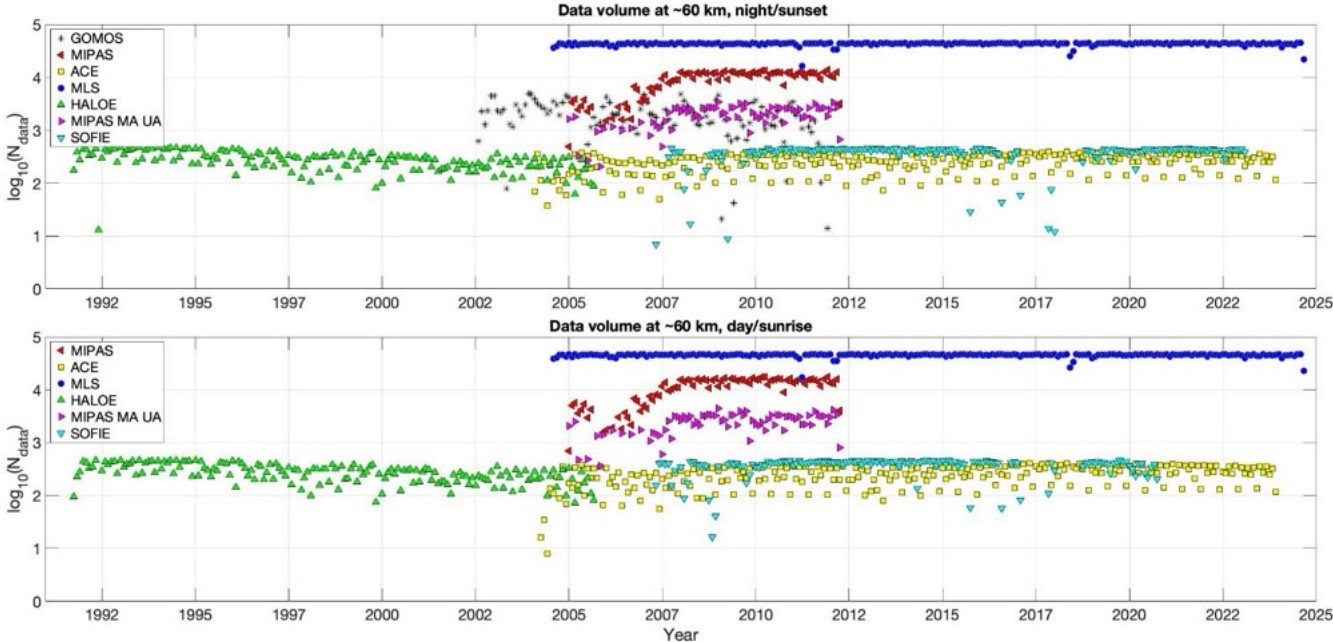

96

**Figure 1: Monthly data volume (logarithm of number of measurements) at 60 km. Top panel: nighttime/sunset measurements, bottom panel: daytime/sunrise measurements.**





## 2.1 HALOE

The Halogen Occultation Experiment (HALOE) was a solar occultation instrument that operated on board the Upper Atmosphere Research Satellite (UARS) from September 1991 until November 2005 (Russell et al., 1993). HALOE typically recorded about 15 sunrise and sunset events per day, between 3 km and 130 km altitude, with a vertical resolution of about 2 km. HALOE was able to cover the entire latitude range from 80°S to 80°N within a time span of about two to six weeks, depending on the time of year.

Here we use Level 2 data obtained with the version 19 processing algorithm. Ozone observations were performed using a broadband spectral channel centered around 9.6 μm. The error estimates account for random noise and altitude-dependent quasi-systematic uncertainties, primarily arising from aerosol correction inaccuracies. These errors are about 5-10% in the middle and upper atmosphere, increasing to around 30% near the 100 hPa level (Bhatt et al., 1999).

## 2.2 GOMOS

The Global Ozone Monitoring by Occultation of Stars (GOMOS) was a stellar occultation instrument that operated on board Envisat (ENVIronmental SATellite) over 2002-2012 (Bertaux et al., 2010; Kyrölä et al., 2010). Ozone profiles are retrieved from the ultraviolet (UV) and visible spectrometer measurements at wavelengths between 250 and 692 nm. The main dataset consists of nighttime ozone profiles (with solar zenith angle larger than 105°), which are retrieved from atmospheric transmittance spectra. For this study, we use GOMOS ozone profiles obtained with the ALGOM2s v1.0 processor (Sofieva et al., 2017a). ALGOM2s is identical to the ESA IPF v6 processor in the stratosphere and MLT but has improved data quality in the UTLS.

GOMOS provides stratospheric and MLT ozone profiles with a vertical resolution of 2 km below 30 km, 3 km above 30 km, with a linear transition between (Tamminen et al., 2010). The vertical resolution of the GOMOS ozone profiles is the same for all occultations due to the Tikhonov-type target-resolution regularization (Kyrölä et al., 2010; Sofieva et al., 2004). The stellar flux recorded by GOMOS, and thus signal-to-noise ratio and precision of retrieved profiles, depends on stellar magnitude and spectral class. The estimated random uncertainty of GOMOS ozone profiles in the MLT is 1-7 % (Tamminen et al., 2010).

GOMOS exploits a self-calibrating measurement principle, therefore high stability of the GOMOS data is expected (Kyrölä et al., 2010). It turned out that it is important to exclude the ozone data from the stars with insufficient UV-flux. In our study, we used the GOMOS data from the HARMOZ dataset, which consists of valid data only.

## 2.3 MIPAS

The Michelson Interferometer for Passive Atmospheric Sounding (MIPAS) was an infrared limb emission spectrometer that was flown on the Envisat platform (Fischer et al., 2008). In 2002-2004, the instrument operated at full spectral resolution.



Due to a failure of the instrument's mirror slide in 2004, operations were suspended for almost 9 months and were resumed in
January 2005 with reduced spectral but improved vertical resolution. These operations continued until the loss of
communications with the ENVISAT platform in April 2012. Most of the time MIPAS observed the 6–68 km altitude range in
its nominal mode (NOM). After 2005, it also pointed to higher altitudes less frequently (about one out of five days) in its
middle atmosphere (MA) and upper atmosphere (UA) measurement modes.
Stratospheric ozone profiles are retrieved from MIPAS/ENVISAT limb emission spectra. In this work, we use version
V8R_O3_261/561/661 ozone data derived with the scientific MIPAS level-2 processor developed by Karlsruhe Institute of
Technology and Instituto de Astrofisica de Andalucía (IMK/IAA). The retrieval is performed via constrained inverse modelling
of limb radiances. A detailed description can be found in von Clarmann et al. (2003, 2009). The data version used in this work
is retrieved from new Level 1 spectra (version V8).  Improvements in the Level 2 retrieval strategy are described in Kiefer et
al. (2023), and López-Puertas et al. (2023) for the NOM and MA/UA modes, respectively.
Due to their different data characteristics, the two MIPAS measurement periods are usually treated as two independent datasets.
Their processing schemes are different, and the vertical resolution of the early MIPAS period is lower than that of the later
period:  5 - 8 km vs. 4 - 6 km for retrieved ozone in the mesosphere from NOM measurements. Therefore, and because of their
short temporal coverage, data of the first measurement period (2002-2004) have not been considered in this study.   The total
random error in the mesosphere from NOM observations is on the order of 8% (at 50 km) to 30% (at 70 km), with a total
systematic error in the same range. For MA and UA observations, the total random error ranges from 3% at 50 km altitude to
20%–30% in the upper mesosphere/lower thermosphere, being larger in daytime than in nighttime. The total systematic error
is rather constant over all altitudes and is in the range of 7 to 10%, including non-LTE (non-Local Thermodynamic
Equilibrium) errors from uncertainties in the collisional and kinetic rate constants and the abundances of atmospheric species
required for the non-LTE modelling (Kiefer et al., 2023; López-Puertas et al., 2023).
**2.4    ACE-FTS**
The Atmospheric Chemistry Experiment - Fourier Transform Spectrometer (ACE-FTS) on-board the Canadian Science
Satellite (SciSat) satellite (Bernath, 2017; Bernath et al., 2005) has been providing the data since February 2004 to present. It
measures from about 85°N to 85°S with complete coverage every 3 months.  The ACE-FTS is a high-spectral-resolution (0.02
cm[-1]) Fourier transform spectrometer observing the 2.2-13 µm range (Bernath et al., 2005). Using solar occultation, it provides
vertical profiles for over 30 atmospheric species.
For each occultation, the ACE-FTS retrieval algorithm determines volume mixing ratio profiles by applying a global
non-linear least-squares fitting technique, matching observed spectra with those produced by a forward model. A full
description of this retrieval processor can be found in Boone et al. (2005). The current version of the ACE-FTS data set used
for HARMOZ is v5.2, as described in (Boone et al., 2023). For ozone, retrieval uses 40 microwindows between 829 and 2673
cm[-1] and accounts for 15 different interfering species and subsidiary isotopologues. The altitude range of the retrieved ozone
profiles is from roughly ~5 km up to about 95 km, with a vertical resolution of approximately 3-4 km.





The ACE-FTS dataset includes quality flags (Sheese et al., 2015). For HARMOZ, data points with flag values > 0 at a
given altitude and any profiles with flags 4–6 were removed. A recent validation study (Sheese et al., 2022) comparing ACE-
FTS ozone with MLS and MIPAS (2004–2012) found an average bias of about 2% from 10–45 km and 0–19% above 46 km.
**2.5    MLS**
The Microwave Limb Sounder (MLS), launched as part of NASA's Aura mission in July 2004, measures millimeter-
and submillimeter-wavelength thermal emission from the limb of Earth's atmosphere (Waters et al., 2006). The Aura MLS
field of view points in the direction of orbital motion and vertically scans the limb in the orbit plane, providing approximately
3500 daily vertical profiles of 16 trace gases, temperature, geopotential height, relative humidity with respect to ice, and cloud
ice water content and path spaced at ~165 km along the orbit track from 82°S to 82°N latitude on every orbit. Aura is in a sun-
synchronous orbit with a 1:45 p.m. local time ascending equator-crossing time; thus, MLS samples a given latitude on either
the ascending (mainly day) or descending (mainly night) portions of each orbit at the same local solar time.
Here we use version 5 Level 2 MLS ozone measurements. The MLS retrieval algorithms, which employ an optimal
estimation approach, are "tomographic" in nature, exploiting information from multiple consecutive limb scans to
simultaneously estimate the two-dimensional (along-track and vertical) state of the atmosphere over multiple Level 2 profiles
(Livesey et al., 2006). The quality and resolution of the version 5 MLS data vary by product and with altitude. For ozone, the
vertical resolution ranges from ~2.5–3 km in most of the stratosphere to 7 km at 0.001 hPa (Livesey et al., 2022). The single-
profile precision degrades from 0.04 ppmv at 100 hPa to 0.1 ppmv at 10 hPa, 0.2 ppmv at 1 hPa, 1.1 ppmv at 0.01 hPa, and
3.4 ppmv at 0.001 hPa. However, precision is generally improved by averaging (with the precision of an average of $N$ profiles
being a factor of the square root of $N$ times smaller than the precision of an individual profile). The systematic uncertainty of
version 5 ozone data is in the range 0.1 to 0.3 ppmv over most of the stratosphere and mesosphere but increases to 0.9 ppmv
at 0.001 hPa.
**2.6    SOFIE**
The Solar Occultation For Ice Experiment (SOFIE) operated on board NASA's Aeronomy of Ice in the Mesosphere
(AIM) satellite from April 2007 until March 2023 (Russell et al., 2009). It conducted measurements across 16 spectral bands,
covering wavelengths from 0.29 to 5.32 micrometers, and provided vertical profiles of temperature and key atmospheric
constituents, including $H_2O$, $O_3$, $CO_2$, $CH_4$, and NO (Hervig et al., 2009a, 2009b; Marshall et al., 2011). The instrument covered
high latitudes of about 65°–85° in both hemispheres with a vertical resolution of about 1.6 km. SOFIE measured 15 sunrise
and sunset events per day between about 20 km and 105 km (Das et al., 2023; Gordley et al., 2009a, b).
Here we use ozone data from SOFIE Level 2 processing (v1.3). The retrievals use radiative transfer simulations and
incorporate the instrument effects. Ozone measurements are made using two broadband filters at 292 nm and 330 nm, with the
former located in a spectral region of strong absorption that offers increased sensitivity in the mesosphere. It enables ozone



measurements over about 50-105 km. Ozone errors result from uncertainties in the observations and the forward model. In the
50 to 100 km altitude range, ozone mixing ratio uncertainty from various random and systematic sources ranges over 3.4–
6.5% (Das et al., 2023).
**3      Preparation and selection of data for merging**
Since the data in the longest dataset, MLS, are retrieved on a pressure grid, pressure was selected as the vertical
coordinate. HALOE ozone profiles are also retrieved on a pressure grid.  MIPAS and ACE-FTS ozone profiles are retrieved
on a geometric altitude grid, but temperature and pressure profiles are also retrieved from the measurements, so the conversion
to a pressure grid is straightforward. GOMOS provides ozone number density profiles on an altitude grid. Since the temperature
and pressure profiles provided in the GOMOS files rely on a combination of ECMWF analyses and the MSIS90 model (Mass
Spectrometer–Incoherent Scatter model; Hedin 1983, 1991) and are therefore not very accurate, we used an altitude-pressure
relationship derived from MIPAS_MA_UA measurements to convert the GOMOS profiles to a pressure grid (see detailed
description in the Supplement).
For creating monthly zonal-mean data from the individual instruments, 10° latitude bands from 80°S to 80°N are used.
The specific challenge for combining ozone data in the mesosphere is that measurements are made at different local times,
while diurnal variations are large in the MLT. Therefore, we first computed monthly zonal-mean ozone profiles for each
illumination condition: daytime (solar zenith angle <90°), nighttime (solar zenith angle >108°), sunset or sunrise. For MLS
and MIPAS, they are daytime and nighttime measurements, for occultation instruments - sunset and sunrise.
For all sensors, the monthly zonal average is computed as the mean of ozone profiles $x_k(z)$, for each illumination
condition:

$$\rho(z) = \frac{1}{N}\sum x_k(z), \qquad (1)$$

where $N$ is the number of measurements. We required $N>10$ in computation of monthly zonal-mean values. The uncertainty
of the monthly mean $\sigma_\rho^2$  can be estimated as the standard error of the mean:

$$\sigma_\rho^2 = \frac{s^2}{N}, \qquad (2)$$

where $s^2 = \langle (x_k - \rho)^2 \rangle$  is the sample variance. In Eq. (2), we used a robust estimator for the sample variance: $s = 0.5(P_{84} -$
$P_{16})$, where $P_{84}$ and $P_{16}$ are the 84$^{th}$ and 16$^{th}$ percentiles of the distribution, respectively.
The monthly zonal-mean wintertime (DJF) ozone distributions for MLS, MIPAS_NOM, MIPAS_MA_UA and
GOMOS are presented in Figure 2. The averaged values were calculated for the period 2005-2011 for these limb
measurements. The top panel shows nighttime conditions, while the bottom panel represents daytime. Across datasets, ozone
distributions are similar, highlighting well-defined features.





The nighttime ozone vertical distribution (top panels in Fig. 2) exhibits three distinct maxima. The primary ozone maximum occurs in the stratosphere, between approximately 30 and 35 km, where ozone mixing ratios reach about 10-11 ppmv (maximum in the tropics). It is produced through a photochemical equilibrium involving oxygen molecules, atomic oxygen, and solar ultraviolet radiation, and is modulated by natural variability and human activities (e.g. ozone-depleting substances, $NO_x$ cycle). The secondary maximum is found around 90-95 km, with the nighttime values as high as the stratospheric maxima (~10 ppmv). The secondary ozone peak represents a short-lived photochemical equilibrium strongly influenced by temperature and by the concentrations of atomic hydrogen and atomic oxygen (Smith and Marsh, 2005). During polar winter, a tertiary ozone maximum of roughly 2-3 ppmv develops near 72 km, peaking close to the polar night terminator. This feature results from reduced concentrations of odd hydrogen during nighttime, decreasing odd-oxygen losses via $HO_x$ catalytic cycles (e.g., Marsh et al., 2001; Sofieva et al., 2009).

In the middle and lower stratosphere, ozone exhibits relatively small diurnal variations because atomic oxygen concentrations are small compared to ozone, and ozone lifetimes exceed one day. As altitude increases into the MLT, diurnal variations become stronger. They are mainly controlled by the daytime photolysis of ozone and its reformation at night through the recombination of atomic and molecular oxygen. The efficiency of ozone production increases with altitude as atomic oxygen becomes more abundant (Brasseur and Solomon, 2005).

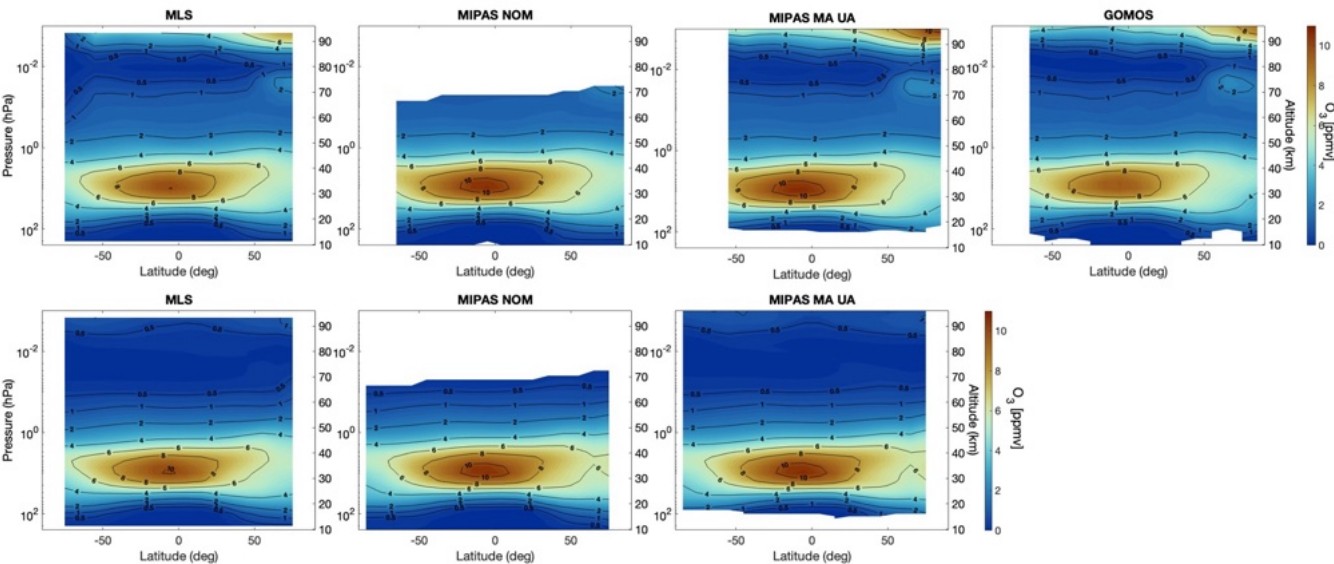

**Figure 2: Wintertime (DJF) zonal-mean ozone profiles for the years 2005-2011 for MLS, MIPAS_NOM, MIPAS_MA_UA and GOMOS (from left to right) for nighttime (top) and daytime (bottom) measurements.**

Figure 3 presents the monthly zonal-mean ozone distributions for winter (DJF) as measured by the occultation instruments ACE, HALOE, and SOFIE. The wintertime means were calculated for the periods 2005-2011 for ACE, 1991-2005 for HALOE, and 2007-2023 for SOFIE. For the most part, similar features are observed as in Fig. 2. The ozone





distributions show good consistency among all datasets, with spatial patterns clearly observed. However, a key difference is the nighttime ozone enhancement above 70 km. Solar occultation observations are limited to periods when the sun is rising or setting, rather than during complete darkness. The pronounced nighttime ozone enhancement in the MLT occurs during full darkness, outside the observation window of solar occultation instruments. The wintertime means from ACE-FTS and HALOE are in good agreement with those previously reported by Smith et al. (2013).

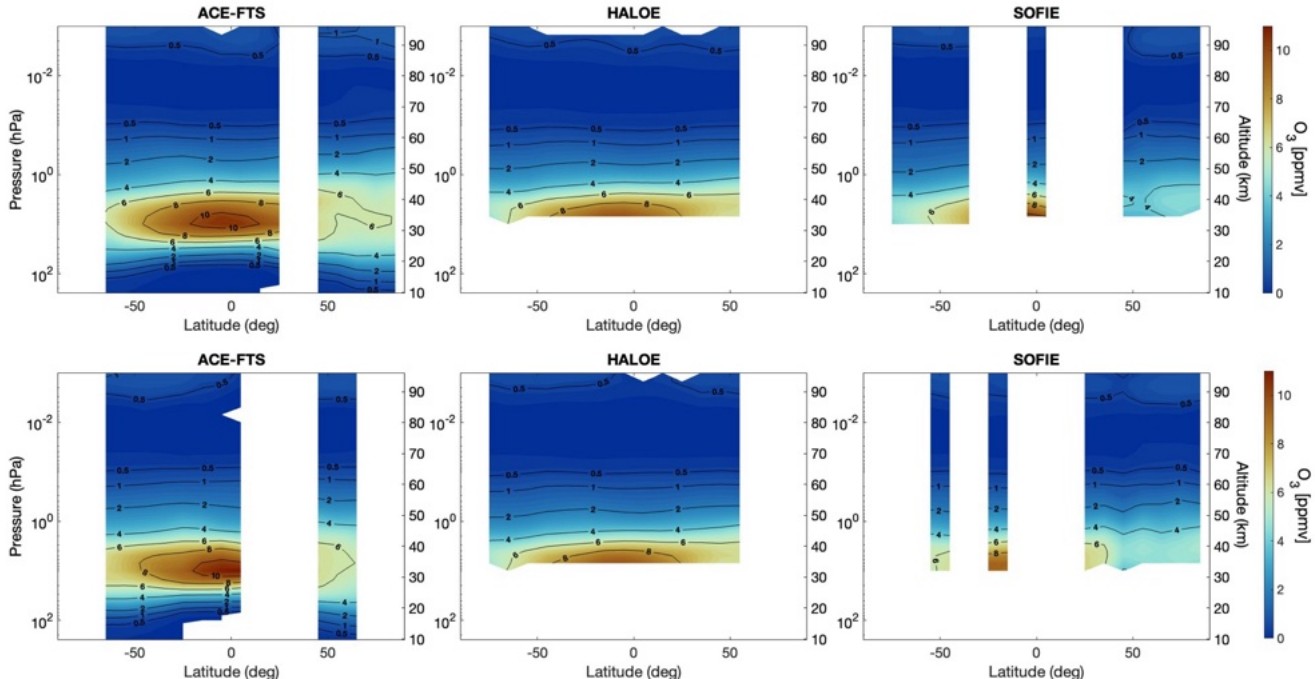

**Figure 3: Wintertime (DJF) zonal-mean ozone profiles for the years 2005-2011 for ACE, 1991-2005 for HALOE and 2007-2023 for SOFIE (from left to right) for sunset (top) and sunrise (bottom) measurements.**

Figure 4 presents an example time series of nighttime ozone at mid-latitudes (30°N-60°N) from four instruments (MLS, MIPAS_NOM, MIPAS_MA_UA, and GOMOS), illustrating the vertical distribution of ozone over time. Temporal variations in ozone are similarly represented across datasets, with key patterns over time clearly visible. Seasonal variations in ozone primarily manifest as an annual cycle in the stratosphere, with a maximum of about 10-11 ppmv in late spring/early summer. In the MLT region, a semi-annual cycle is observed, with ozone peaks of about 10-12 ppmv occurring around March-May and September-November.






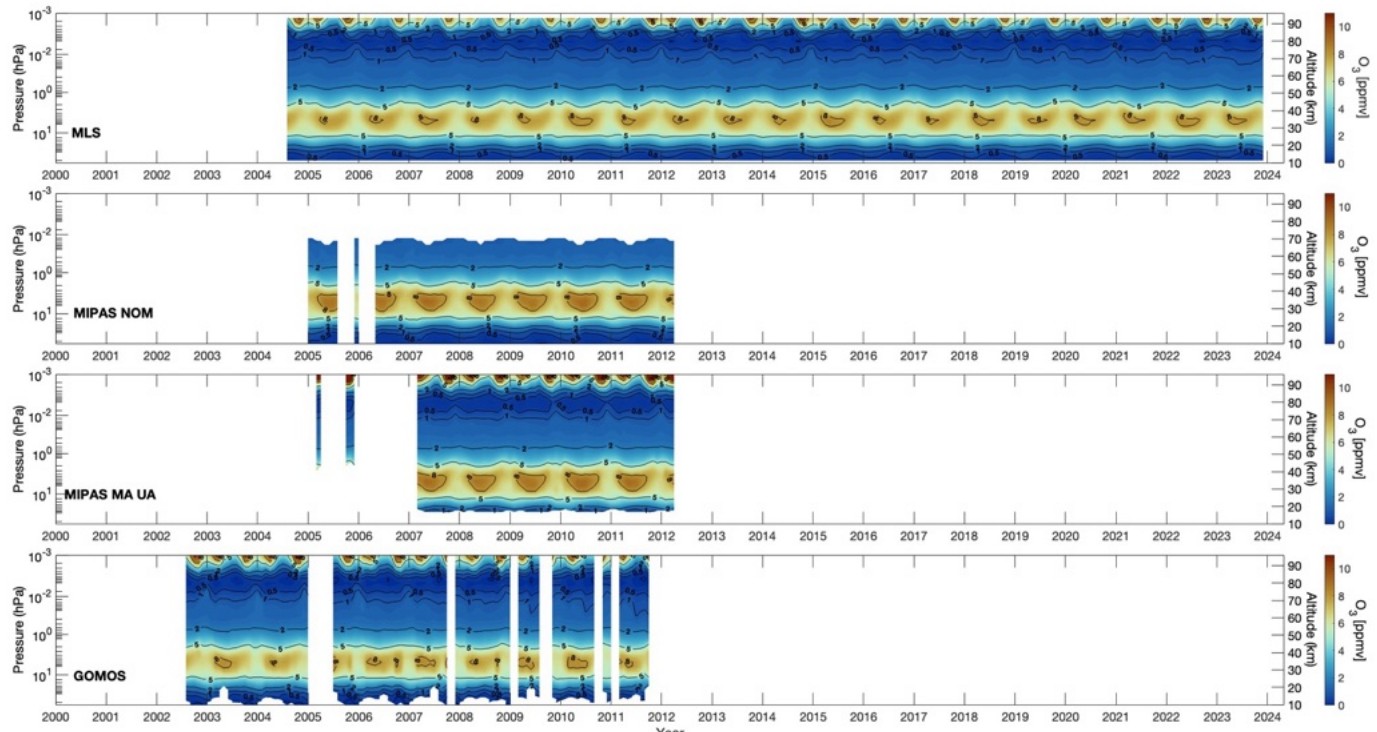


**Figure 4: Time series of nighttime ozone at mid-latitudes (30°N–60°N) from MLS, MIPAS_NOM, MIPAS_MA_UA, and GOMOS**
**(from top to bottom).**
## 4    The merging procedure

764        The merging procedure in general is like that used by Sofieva et al. (2017b, 2023) but is adapted for the MLT region

by considering the strong diurnal cycle of ozone.  Below we present the details of the merging procedure. It consists of three
main steps: (i) evaluation of deseasonalized anomalies, (ii) data pre-merging, and (iii) final merging of pre-merged datasets
from individual instruments.
### 4.1    Deseasonalized anomalies and seasonal cycle

769        For each instrument, illumination condition, latitude band and altitude level, the deseasonalized anomalies are

computed as:
$$\Delta(t_i) = \frac{\rho(t_i) - \rho_m}{\rho_m}, \qquad (3)$$



where $\rho(t_i)$ is the monthly mean value at a certain altitude and latitude band corresponding to time $t_i$, and $\rho_m$ is the mean value for the corresponding month $m$, i.e. $\rho_m = \frac{1}{N_m}\sum_{j=1}^{N_m}\rho_j$, $N_m$ being the number of monthly mean values $\rho_j$ in a given month $m$ available from all years. The uncertainty of the seasonal cycle value $\rho_m$ for each month $m$ is evaluated from uncertainties of individual monthly mean values $\sigma_{\rho,j}$:

$$\sigma_m^2 = \frac{1}{N_m^2}\sum_{j=1}^{N_m}\sigma_{\rho,j}^2 , \qquad (4)$$

For GOMOS, MIPAS (NOM and MA_UA) and MLS, the seasonal cycle is evaluated using the period 2005-2011. For HALOE and SOFIE, their full operation periods are used for evaluation of seasonal cycles. For ACE-FTS, the years 2005-2018 are used for evaluation of the seasonal cycle. The uncertainty of deseasonalized anomalies is evaluated using error propagation.

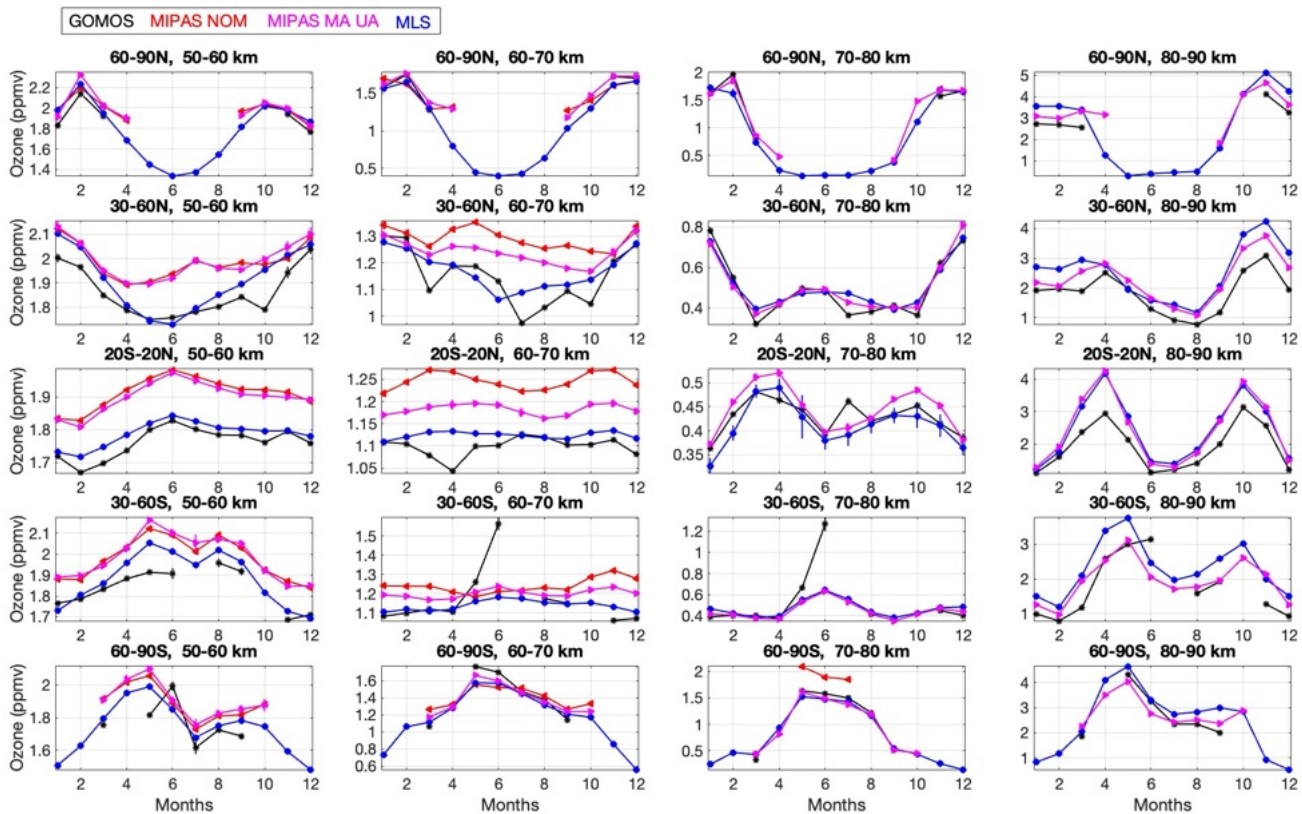

**Figure 5: Ozone seasonal cycle for nighttime measurements in latitude bands 60°N-90°N, 30°N-60°N, 20°S-20°N, 30°S-60°S and 60°S-90°S (from top to bottom) for altitudes 50-60 km, 60-70 km, 70-80 km and 80-90 km (from left to right). The seasonal cycles in the indicated zones are computed as the mean of seasonal cycles in 10° latitude bands.**





Figure 5 illustrates the seasonal cycle of nighttime ozone measurements for different latitude bands: 60°N-90°N,
30°N-60°N, 20°S-20°N, 30°S-60°S, and 60°S-90°S (from top to bottom), across four altitude ranges: 50-60 km, 60-70 km,
70-80 km, and 80-90 km (from left to right). The semi-annual cycle in the MLT is clearly visible, with ozone concentrations
exhibiting two peaks per year during equinoxes.
Agreement among datasets is generally good, though some discrepancies between instruments exist, consistent with
earlier validation and intercomparison results. Additionally, GOMOS shows some biases, which can be attributed to its coarser
temporal and horizontal sampling, compared to MLS and MIPAS.

## 4.2     Pre-merging

Ozone in the mesosphere has strong diurnal variations, as was mentioned above and illustrated in Figures 2, 3 and
Figure 6 (top). However, the deseasonalized anomalies in different illumination conditions are nearly identical for the
instruments with dense sampling (such as MIPAS and MLS; see the case for MIPAS_MA_UA in Figure 6, bottom). This
allows us to consider the weighted mean of daytime and nighttime deseasonalized anomalies as the aggregated (we will call
them "pre-merged" hereafter) deseasonalized anomaly from each instrument. The weights are inversely proportional to the
estimated uncertainties of the deseasonalized anomalies. As shown in Figure 7, the daytime, nighttime, and weighted-mean
deseasonalized anomalies are seen to be similar over the full altitude range of MIPAS_MA_UA measurements. For occultation
instruments, the weighted mean of sunset and sunrise anomalies is considered as a pre-merged anomaly.

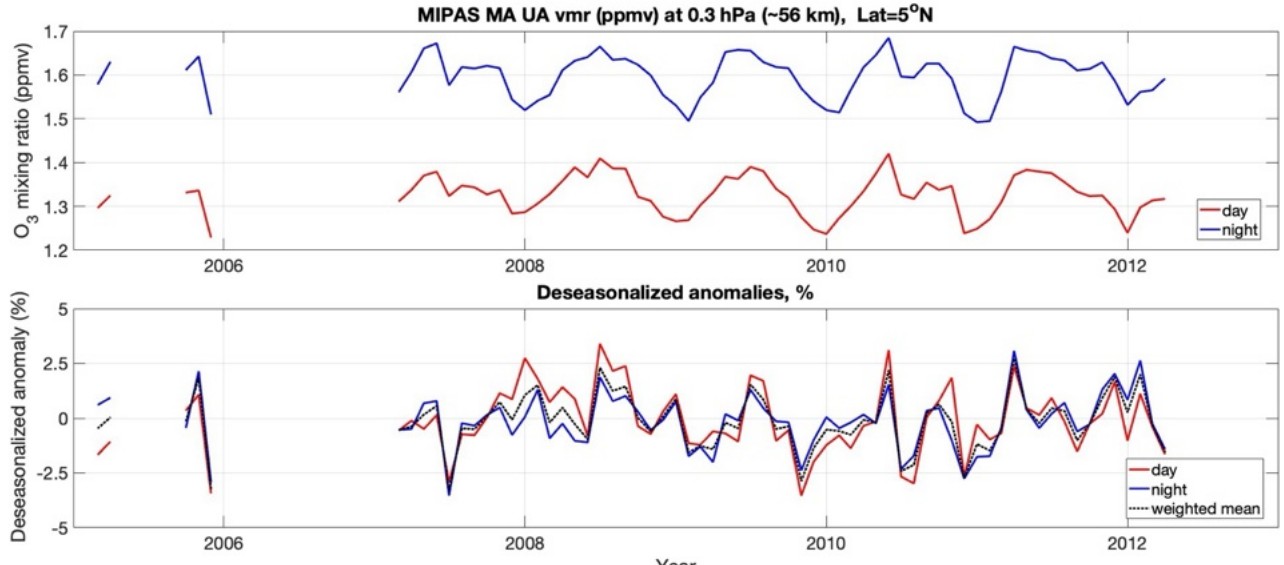


**Figure 6: Top: ozone mixing ratio at 56 km for MIPAS_MA_UA daytime and nighttime observations in latitude zone 0-10°N.**
**Bottom: the corresponding deseasonalized anomalies and the weighted mean anomaly (see text for explanation).**

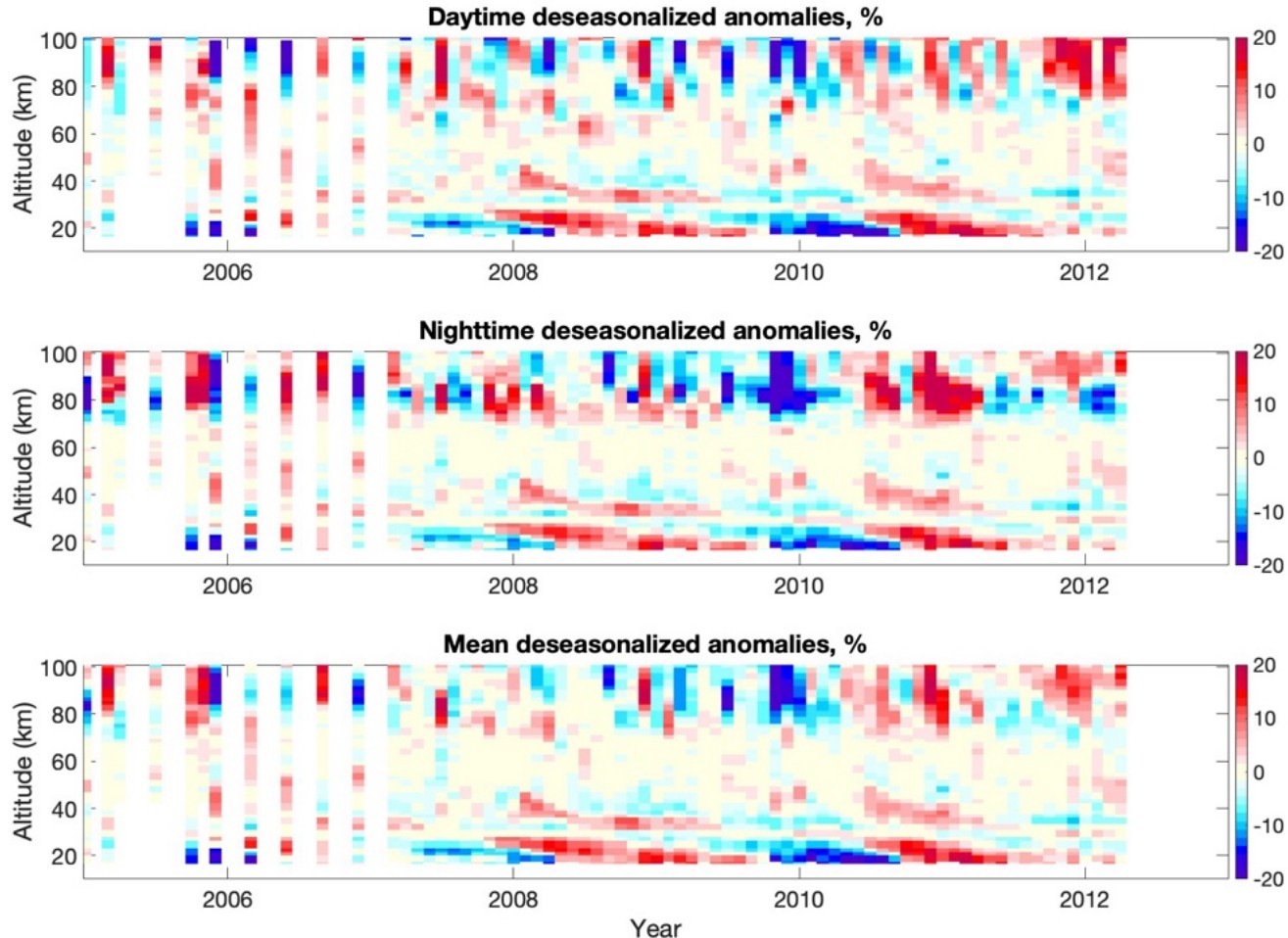

**Figure 7: MIPAS_MA_UA deseasonalized anomalies at 0-10°N. Top: daytime anomalies, center: nighttime anomalies, bottom: the weighted mean of daytime and nighttime anomalies.**

### 4.3 The merging procedure

The merging procedure is like that used for the SAGE-CCI-OMPS+ merged dataset (Sofieva et al., 2017b, 2023). First, we take the median of deseasonalized anomalies from all pre-merged datasets except for HALOE and SOFIE. Then HALOE and SOFIE pre-merged anomalies, for which the seasonal cycle is computed using different time periods, are offset to this median anomaly. These offsets are usually very small (see Figure 8, which illustrated the merging procedure). Finally, the median of all aligned anomalies is computed. The merging procedure is performed for each latitude zone and for each pressure level. An example of vertical profiles of pre-merged deseasonalized anomalies from individual datasets and the merged anomaly is shown in Figure 9 for the latitude zone 60°S-70°S. The altitude coverage slightly varies between the instruments. The best spatio-temporal coverage is attained after 2004.





When performing data merging, we also analysed the deviations of individual pre-merged anomalies from the merged anomaly, to detect drifts or strong deviations. Such a comparison is shown in Figure 10. No clear drifts were observed; the pre-merged anomalies from different instruments are in good agreement with each other.

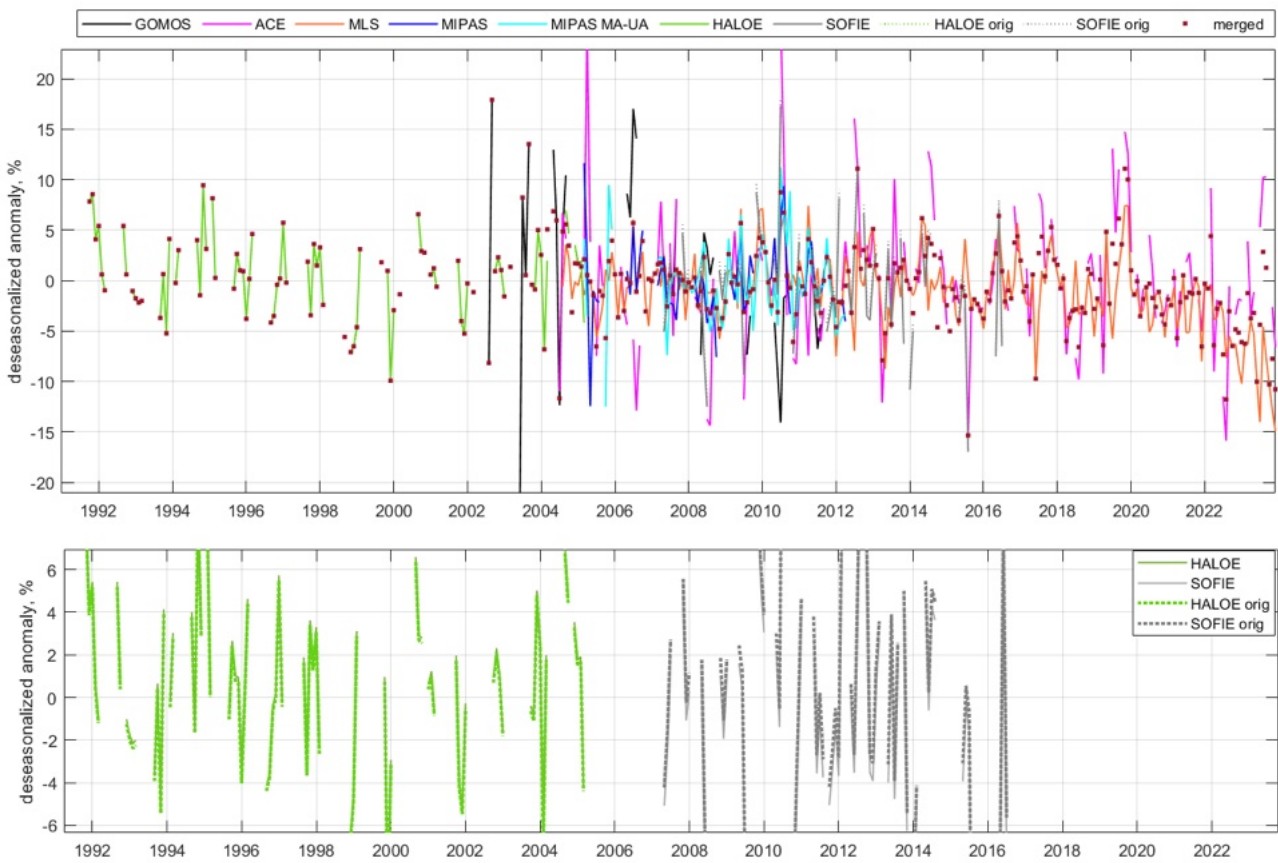

**Figure 8: Illustration of the merging procedure using the data at 60°S-70°S, 0.05hPa. Color lines: individual pre-merged deseasonalized anomalies for different instruments (see legend). For HALOE and SOFIE, the original anomalies are shown by dotted lines, while offset anomalies are shown by solid lines. The final merged anomaly is shown by dark red dots. Top: all data, bottom: a zoom on HALOE and SOFIE anomalies for visualization of small offsets.**



**Figure 9: An example of vertical profiles of pre-merged deseasonalized anomalies from individual datasets and the final merged anomaly. The data are from the latitude zone 60°S-70°S.**



**Figure 10: Example of deviations of individual deseasonalized anomalies from the merged anomaly. The data are from the latitude zone 60°S-70°S.**

The uncertainties of the merged ozone anomalies are evaluated as proposed by Sofieva et al., (2017b, 2023):

$$\sigma_{\Delta,merged} = min\left(\sigma_{\Delta,j_{med}}, \sqrt{\frac{1}{N}\sum_{j=1}^{N}\sigma_{\Delta,j}^2 + \frac{1}{N^2}\sum_{j=1}^{N}(\Delta_j - \Delta_{merged})^2}\right), \quad (5)$$

where $\sigma_{\Delta,j_{med}}$ is the anomaly uncertainty of the instrument corresponding to the median value. Thus, the uncertainty of merged anomalies depends on the agreement between the deseasonalized anomalies used for the computation of the median values, and it is smaller than $\sigma_{\Delta,j_{med}}$ if several instruments report a similar anomaly. The typical uncertainties of the merged deseasonalized anomalies are shown in Figure 11 for several latitude bins. In general, the estimated uncertainties are below 2%, except in the mesopause region, where they increase to 5-10%. Before 2002, when the merged record consists solely of HALOE measurements, the estimated uncertainties above 75 km are 5-10% and <2% below 70 km.

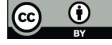





**Figure 11: Examples of uncertainties (in %) of the merged deseasonalized anomalies. The latitude bins are indicated in the panels.**

## 5    The merged METEOR-O3 dataset of ozone profiles

The deseasonalized anomalies can be directly used for ozone trend analyses. For other applications, we also created merged ozone mixing ratio profiles. For this, we computed the merged seasonal cycle as the mean of seasonal cycles from MLS, MIPAS_NOM and MIPAS_MA_UA, for daytime and nighttime illumination conditions. These datasets are selected because of their dense sampling, and for maximum spatial coverage.    Then this seasonal cycle is applied to the merged deseasonalized anomalies (Eq. 3). Thus, in addition to deseasonalized anomalies, nighttime and daytime ozone mixing ratio profiles and their uncertainty are provided in the merged dataset. Examples of the merged METEOR-O3 mixing ratio profiles are shown in Figure 12.



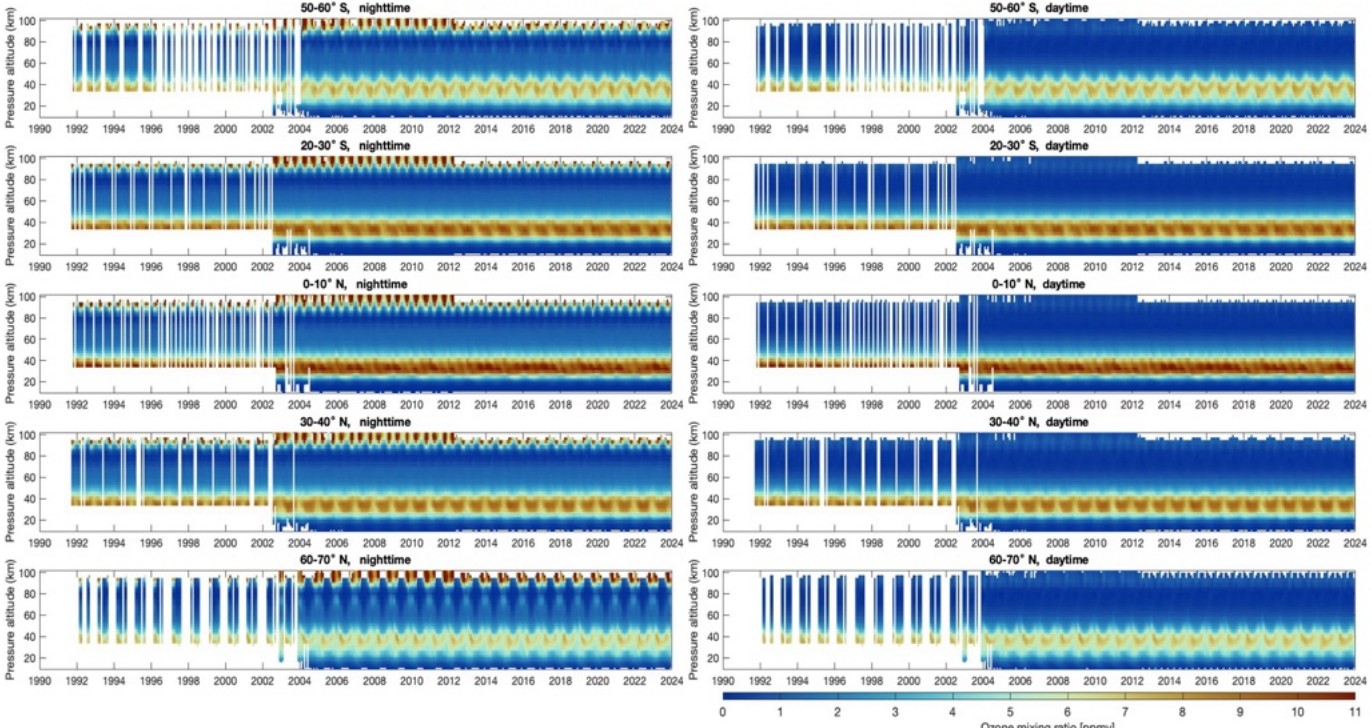


**Figure 12: Examples of the merged METEOR-O3 nighttime (left) and daytime (right) time series of ozone mixing ratio profiles. The**
**latitude bins are indicated in the panels.**

## 6    Analyses of trends in the upper atmosphere

54       The merged deseasonalized ozone anomalies are suitable for direct use in ozone trend assessments. For this analysis, we

apply a multiple linear regression (MLR) to the METEOR-O3 dataset:
$$O_3(t) = PWLT(t, t_0) + q_1 QBO_{30}(t)) + q_2 QBO_{50}(t)) + sF_{10.7}(t)) + dENSO(t), \tag{6}$$
where $PWLT(t, t_0)$ is a piecewise linear term (constant and a hockey-stick trend with the turnaround point in 1997), $QBO_{30}(t)$
and $QBO_{50}(t)$ are two quasi-biennial oscillation (QBO) proxies (30 hPa and 50 hPa equatorial winds;
http://www.cpc.ncep.noaa.gov/data/indices/), $F_{10.7}(t)$ is the monthly average solar 10.7 cm radio flux
(https://www.spaceweather.gc.ca/forecast-prevision/solar-solaire/solarflux/ sx-5-en.php), and $ENSO(t)$ is the 2-month lagged
El Niño–Southern Oscillation (ENSO) proxy (https://www.esrl.noaa.gov/psd/enso/mei/data/meiv2.data). Similar formulations
have been used in previous ozone trend studies (e.g., Bourassa et al., 2014; Kyrölä et al., 2013; Sofieva et al., 2017b).
Uncertainties are derived from the residuals of the regression fits, and autocorrelation is corrected in both steps using the
Cochrane-Orcutt transformation (Cochrane and Orcutt, 1949).





For comparison, the regression analysis was performed in two ways. In the first approach, the MLR was applied to the entire merged dataset covering the period 1991-2023. In this case, the bottom altitude range was limited to the coverage of HALOE observations, starting at approximately 37 km and running up to 93 km. In the second approach, the MLR was applied only to the period with the best spatio-temporal coverage (2004-2023), extending from about 22 km upward.

Figure 13 displays the trend results for the recovery period (1997-2023) and for the 2004-2023 period. The trends are very similar for both considered periods. Stratospheric ozone trends are in very good agreement with previous ozone assessments (Godin-Beekmann et al., 2022; Petropavlovskikh et al., 2019). The results indicate continued recovery of ozone in the upper stratosphere, with positive values of about 1-2% per decade between 35 and 45 km, particularly pronounced at middle and high latitudes in both hemispheres.

In contrast, MLT ozone exhibits negative trends, reaching about 1-3% per decade over 60-80 km. The strongest decreases, of approximately 8-12% per decade, occur between 80 and 90 km. Previous analysis based on 10 years (2002-2012) of SABER (Sounding of the Atmosphere using Broadband Emission Radiometry) data confined to lower and middle latitudes (48°S-48°N) revealed a marginally positive not-significant ozone trend at altitudes of 60-80 km and a strong negative trend up to 10% per decade near 85-100 km. Positive/negative ozone trends were accompanied by a cooling trend of about 3 K per decade (Huang et al., 2014). Regionally, Bizuneh et al. (2022), using 16 years of SABER data over low latitudes (5°N-15°N), have shown negative temperature and ozone trends (-0.85 K per decade and -0.12 ppmv per decade) in the lower mesosphere (60-80 km) but positive trends (1.25 K per decade and 0.27 ppmv per decade) in the upper mesosphere/lower thermosphere (85-100 km). Alongside natural drivers, the authors attributed these patterns to the influence of increasing greenhouse gas concentrations. Furthermore, rising $H_2O$ levels in the mesosphere and lower thermosphere can enhance $HO_x$ production, which accelerates ozone loss and may further contribute to the observed negative $O_3$ trends.

In addition to the relative (% per decade) analysis, we have also derived trends based on absolute (ppmv) ozone values (Supplementary Figure S4). These trends are predominantly negative between 60 and 90 km, in qualitative agreement with regional studies (Bizuneh et al., 2022). In the 60-80 km range, trends vary from about -0.01/-0.03 ppmv per decade in tropical and mid-latitudes to -0.05/-0.1 ppmv per decade in polar regions. Between about 80 and 90 km, negative trends up to -0.2 ppmv per decade are observed. Above 90 km, ozone trends exhibit a transition toward weaker negative or even positive values, reaching 0.1-0.3 ppmv per decade at northern mid-latitudes.

The trends in ozone and temperature in the MLT are closely linked to radiative and chemical processes. The well-established ozone and temperature correlation is negative between about 30 and 75 km, where photochemical reactions dominate, and positive above ~80 km, where dynamical processes become stronger (Brasseur and Solomon, 2005). This relation was confirmed using SABER data, showing that ozone and temperature are positively correlated in the upper mesosphere/lower thermosphere but negatively correlated in the lower mesosphere, with the 60-80 km region representing a transition zone where correlations become weak or inconsistent depending on local time, solar activity, and dynamical variability (Huang et al., 2014, 2016). As temperature trends in the region of our analysis are negative (Laštovička, 2023), the results in Fig. 13 are in line with those findings.



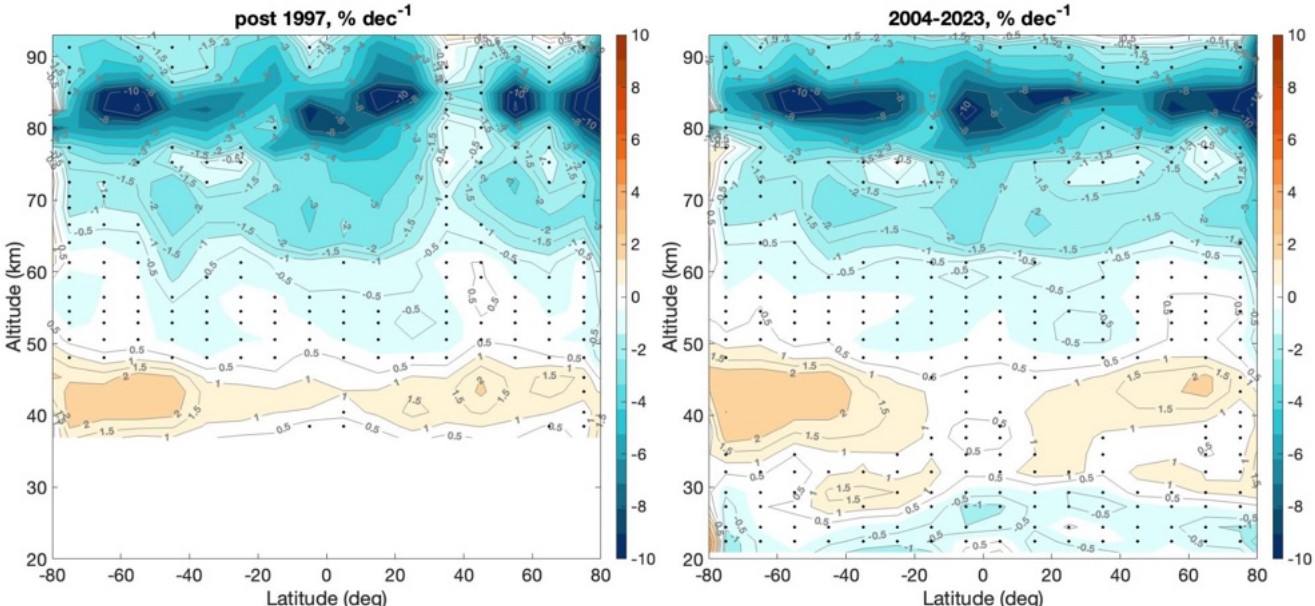

**Figure 13: Latitude-altitude variation of ozone trends derived from the METEOR-O₃ dataset. The left panel shows post-recovery trends (1997-2923), and the right panel shows trends for 2004-2023. The black dots indicate trends that are not statistically significant at the 95% confidence level. Trends are given in % per decade.**

To investigate the seasonal dependence of ozone trends, we applied a two-step multiple regression technique like that described by Szelag et al. (2020). In the first step, natural cycles (solar, QBO, and ENSO) were estimated and removed from the data using the traditional MLR formulation given above (Eq. 6). The regression was performed using data from each (three-month) season separately. In the second step, the residual time series (after removal of natural variability) was used to estimate linear trends for the recovery period (2000-2023) using a simple linear regression. This two-step approach provides a robust estimation of seasonal trends by ensuring sufficient data for detecting natural cycles and by fitting linear trends only within periods where ozone changes are approximately linear.

Variations of ozone trends over the period 2000-2023 for each latitude and altitude are shown for each season separately in Figure 14. In the upper stratosphere, ozone trends are positive throughout all seasons and across most latitudes, consistent with previous studies (Szelag et al., 2020). The strongest positive trends, up to about 2-4% per decade, are observed at middle and high latitudes.

In the lowermost mesosphere (50-60 km), ozone trends are generally close to zero, with most not statistically significant, suggesting little long-term change in this region. Between approximately 60 and 80 km, negative trends (about 2-4% per decade) dominate, indicating a persistent decrease in mesospheric ozone. The largest negative values, reaching up to 15-20% per decade, occur between 80 and 90 km, particularly in the Northern Hemisphere during boreal summer (JJA) and in the Southern Hemisphere during austral summer (DJF) where the ozone absolute values are very low. During the equinoctial





·19 seasons, negative ozone trends peak at mid-latitudes and in the tropics. In some seasons and latitude bands, however, trends

·20 become positive, for example during local winter between 75-80 km and 60-80° latitudes.

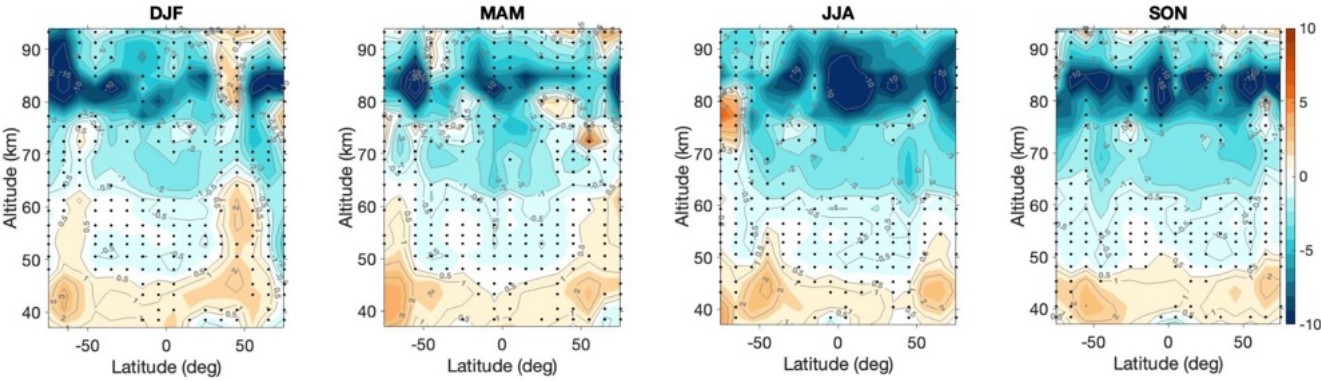

·21

·22 **Figure 14: Latitude–altitude distribution of ozone trends derived from the METEOR-O₃ dataset for each season (DJF, MAM, JJA,**

·23 **SON) over the period 2000-2023. The black dots denote trends that are not statistically significant at the 95% confidence level.**

·24 **Trends are given in % per decade**.

·25 **7   Summary**

·26     We have developed a new merged dataset of mesospheric and lower thermospheric ozone, METEOR-O₃, constructed

·27 from multiple limb emission and occultation satellite instruments (HALOE, MLS, ACE-FTS, MIPAS, GOMOS, and SOFIE).

·28 The merging procedure, adapted from previous methods for stratospheric datasets, accounts for the strong diurnal variability

·29 of ozone in the mesosphere. The dataset provides monthly zonal-mean ozone anomalies from 1991 to 2023, with 10° latitude

·30 resolution and vertical coverage from approximately 22 km up to 100 km.

·31     The merged dataset shows excellent internal consistency among instruments and provides robust coverage of the middle

·32 and upper atmosphere, with typical uncertainties below 2% and slightly higher uncertainties (5-10%) near the mesopause. The

·33 deseasonalized anomalies derived from METEOR-O₃ are well suited for direct use in trend analyses.

·34     Ozone trend evaluation was carried out using two complementary regression methods. The one-step MLR analysis was

·35 applied to the full merged dataset (1991-2023) and to the period of best spatio-temporal coverage (2004-2023). For the full-

·36 period analysis, only post-recovery (post-1997) trends were presented. The results show positive trends of about 1-2% per

·37 decade in the upper stratosphere (35-45 km), consistent with the ongoing ozone recovery observed in earlier studies (Godin-

·38 Beekmann et al., 2022; Petropavlovskikh et al., 2019). In the mesosphere, ozone trends are negative, with values of

·39 approximately 1-3% per decade in the 60-80 km region and up to 8-10% per decade between 80 and 90 km. While previous

·40 SABER-based studies were temporally and regionally limited (e.g., Bizuneh et al., 2022; Huang et al., 2014), the present



analysis extends to a global scale, providing for the first time a comprehensive assessment of long-term mesospheric/lower
thermospheric ozone variability.
The METEOR-O$_3$ dataset enables global evaluation of long-term changes and seasonal variations. It offers a valuable
resource for model validation and for improving understanding of upper atmospheric processes.
**Code and data availability**
The merged METEOR ozone dataset link will be provided with manuscript release.

**Author contributions**
MES and VFS designed the study, performed the analyses and wrote the manuscript. All authors provided data and
contributed to the analyses and writing of the paper.

**Competing interests.**
At least one of the (co-)authors is a member of the editorial board of Atmospheric Chemistry and Physics.

**Acknowledgments**
The work at Finnish Meteorological Institute has been performed within the framework of the ESA METEOR project. Work
at the Jet Propulsion Laboratory, California Institute of Technology, was carried out under a contract with the National
Aeronautics and Space Administration (80NM0018D0004).

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
