# Peer review of "Evaluation of ozone trends in the mesosphere/lower thermosphere"

_EGUsphere, 2025_

## Referee Comment (RC2)

**General comments**

This manuscript describes a new merged dataset, METEOR-$O_3$, which can be used for long term ozone trend analysis in the mesosphere/lower thermosphere. Profile observations of this portion of the atmosphere are crucial, but often overlooked, and this work goes far towards addressing existing gaps. The manuscript begins with a description of the procedure used to create the merged dataset and ends with a trend analysis using the final product.

Overall, the methodology described appears to be consistent with that used for previous merged datasets. However, I have a few important concerns related to the methodology (see #1.1-1.3 below) which should be addressed. Besides those issues, there are several points where the presentation of the methodology needs major clarifications (see #1.4-1.8). I am also providing many minor comments and technical corrections (see #2).

**1. Specific comments:**

1.1. Why start at 22 km if this is a MLT dataset? If the full range is provided, isn't this more of a stratosphere-to-LT dataset? Calling it a MLT dataset may be underselling its value, unless there are considerations when utilizing data from the stratosphere?

1.2. The merging procedure relies on median values. Given that medians are calculated from the deseasonalized anomalies, I would assume these are each effectively close to 0? However, medians are then matched across varying time periods. I think this procedure may be problematic given the region where the negative trends are found later in the text (e.g. Fig. 13) is at altitudes where the anomalies are also close to 0. To reduce artefacts of the merge, I would expect medians to be aligned over overlapping time periods only. Perhaps further clarity on the description of the procedure in Section 4.3 would address this choice of methodology.

1.3. I would expect the ozone profiles in Section 5 to be calculated using all datasets, same as the merged anomalies. There is no clear reason why all datasets couldn't be used for the merged seasonal cycle. Perhaps related, the regions of significance in Fig. S4 which provide ozone trends in ppmv vary notably from those in Fig. 13. This suggests that the ppmv profiles and their uncertainties are impacted by the methodology choices.

1.4. Table 1:
   1.4.1. The caption in Table 1 states "Information about the datasets used in the merged dataset", however, this table only lists information about the instruments. It is important to distinguish between full instrument dates, vertical range, etc. and those selected for inclusion when creating the datasets in this study. The reader is likely to be most interested to know the ranges used for METEROR-$O_3$. A good example is MIPAS, which appears to be split into two separate time ranges (with only the latter used) and two different modes.
   1.4.2. Similarly, I suggest starting Table 1 with a column for dataset name, as these are never defined (i.e. MLS, MIPAS_NOM, MIPAS_MA_UA, ...). Following this, the term "dataset" should be used throughout the text, rather than "instrument" or "sensors", when discussing the data used for the merge.
   1.4.3. The column called "Local time/equator crossing time" is hard to reconcile with the illumination conditions referred to later. I would recommend changing this

column to something like, "Illumination conditions" and listing "daytime, nighttime, sunset, sunrise" accordingly. This would clarify later text such as L. 208-209 where GOMOS is not listed.

    1.4.4. There are numerous discrepancies between the values listed in Table 1 and those in the text from Section 2.

1.5. The purpose of the text starting at L. 219 (and Figs 2-4) should be clarified. These appear to be a series of validations to ensure the datasets generally match with each other. But there is also a large portion of text providing background on the general distribution of ozone. Is the goal of Fig. 2 and 3 to ensure that each dataset shows the "key patterns" in ozone?

    1.5.1. I would recommend adding a separate section header to indicate the goals of this text/figures or add more clarify in the text.

    1.5.2. Similarly, the datasets within Fig. 2 are limited to the same time period, but this is not the case for Fig. 3 (understandable, given the time period of ACE). It is unclear whether these choices are made for intercomparison purposes, or to match the years selected for the merge, or a mix of both. This makes it hard to know what the reader is meant to get out of these figures.

1.6. Statistical equations:

    1.6.1. Many variables used are not explicitly defined in the text. This is crucial for readers to understand the methodology. Please provide all definitions across each equation (e.g. $\rho(z)$, $k$, $z$, $\Delta(t_i)$, $\sigma_\Delta^2$, ...)

    1.6.2. Eq. 2:

        1.6.2.1. $s$ appears to be defined two different ways. This should be clarified. If the robust estimator is used, a citation should be provided.

        1.6.2.2. Is $\sigma_\rho^2$ as written here used anywhere?

    1.6.3. Eq. 3:

        1.6.3.1. $\rho(z)$ is the monthly zonal average term from Eq. 1 – is this not the same as $\rho_m$? And if they are the same, why is $\rho_m$ defined differently?

        1.6.3.2. Is it correct to interpret $t_i$ as time in years? If so, this should be stated. Also, what does $i$ mean here, and is it necessary?

    1.6.4. Eq. 4:

        1.6.4.1. The uncertainty of $\rho_m$ should be the same as $\sigma_\rho^2$ right? If so, why is it defined differently in Eq. 4, with $\sigma_\rho^2$ now corresponding to the uncertainty for a single year's monthly mean?

    1.6.5. Generally, the introduction of $\rho_m$ and $\Delta(t_i)$ is confusing. Is $\rho_m$ what is shown in Fig. 5? And this is the same as $\rho(z)$? It may be helpful to show Fig. 5 along with the definition of $\rho_m$ and its uncertainty and afterward define $\Delta(t_i)$ (i.e. reverse the order of Eq. 3 and 4 and more explicitly indicate that Fig. 5 shows $\rho_m$)

        1.6.5.1. Unless I'm missing something, the data shown in Fig. 5 is essentially the same as that from Fig. 2-4, with different axes/selections? I suggest reorganizing all sections between L. 210 and L. 292 to make this section clearer and easier to follow.

    1.6.6. Eq. 6:

        1.6.6.1. Extra ")" in several terms

1.7. Dataset time selections:

1.7.1. According to Table 1, Section 2.6, and Figures 1 and 3, SOFIE data runs from 2007-2023. However, in Figures 8, 9, and 10, SOFIE data appears to run from 2007-2016 within the merged dataset. This is not explained anywhere in the text.

1.7.2. For the seasonal cycle calculations, why is 2005-2011 used for MLS, when 2005-2018 is used for ACE-FTS?

1.7.3. When taking the median of pre-merged datasets (Section 4.3), are the full time periods used? (i.e. for MLS, is 2005-2011 used, or 2005-2023?) The seasonal cycle for ACE_FTS is also calculated "using a different time period" – why? Even with Fig. 8, it's quite difficult to fully understand the procedure here.

1.8. Trends in the MLT:

1.8.1. As far as I can tell, regions of negative trend occur where ozone concentrations are very low and represent very small changes. However, these results do generally appear significant and seem useful to report. Including S4 for context following Fig. 13 would help the reader gauge the magnitude of the detected changes in ppmv.

1.8.2. I expected to see trends at either the secondary or tertiary maxima. However, the strongest region of negative trend in Fig. 13 appears to be just below the secondary maximum, although this is unclear. Given that these maxima and their drivers were introduced on page 9, I would expect the discussion of the trends in Fig. 13 to refer to those locations specifically.

**2. Minor comments and technical corrections:**

2.1. METEOR-$O_3$ vs. METEOR-O**3** inconsistency throughout

2.2. Why does METEOR-$O_3$ end in 2023, when some instruments are still operational?

2.3. Introduction:

2.3.1. Paragraph 1: I suggest defining the altitude range of the MLT in the first sentence. More citations would support the statements in the first two sentences. "Over the last years" → "Over the last few years"?

2.3.2. Paragraph 2: Statements about "cooling" are very general. Where is this cooling found? And should this be discussed as a "trend" rather than a "signal"?

2.3.3. Paragraph 4: More citations are needed to support the statements in the sentences beginning on L. 54 and L. 59.

2.4. L. 79: "we used the limb and occultation instruments that provide data in the mesosphere and the lower thermosphere" → Limb and occultation instruments that observe ozone specifically

2.5. L. 83: "Note that for some instruments, the selected period is shorter than their full operational period" → with Table 1 as-is, it is hard to differentiate between operational and selected periods, and why these vary.

2.6. L. 84-86: What is the relevance of instruments with different illumination conditions having comparable numbers of profiles?

2.7. Table 1: MA UA and NOM not defined in Table 1 caption. Neither is the notation for "20(40)"

2.8. L. 116: Please define UTLS

2.9. "On-board" vs. "on board" inconsistency throughout

2.10. L. 164: "0-19 %" and the citation for this statement should be verified

2.11. Section 2.5: Discussing uncertainty for MLS as % rather than ppmv would be more consistent with how uncertainty is discussed in the other datasets.

2.12. L. 191: What are "the instrument effects"?

2.13. L. 197: MLS is not the "longest dataset", ACE-FTS is

2.14. DJF should be indicated as "boreal wintertime" rather than "wintertime" throughout, as this dataset covers both hemispheres

2.15. All figures: The datasets labelled in the legend should match the dataset names throughout (e.g. MIPAS_MA_UA)

2.16. L. 225-227: Citation needed

2.17. L. 252: "mid-latitudes" → "Northern Hemisphere mid-latitudes"?

2.18. L. 291: "earlier validation and intercomparison results" → specifically which?

2.19. L. 291-292: Regarding the explanation for GOMOS's bias, it may be helpful to point the reader back to Fig. 1 and Section 2.2.

2.20. Fig. 6: Legends should say "daytime" and "nighttime" for consistency. The vertical range on the y-axis could be reduced in the bottom panel.

2.21. Section 4.2: Fig. 6 and 7 are not formally introduced (especially Fig. 6 which is first referenced regarding ozone's diurnal cycle in L. 295) which makes it hard to tell the goals of these figures. Is it a true statement these show an example of the pre-merge from MIPAS_MA_UA from a couple of viewpoints? And can it be assumed that this was a validation procedure performed across all datasets and the same conclusions held? A statement to this effect would be important to add.

2.22. L. 312: "illustrated" → "illustrates"

2.23. Fig. 9, S3, S4: Missing unit on the colorbars

2.24. L. 346-347: "merged deseasonalized anomalies (Eq. 3)" → I'm not sure that Eq. 3 is the correct reference, as the deseasonalized anomalies have not been merged in that equation.

2.25. L. 374: "60-80 km" → "60-90 km"?

2.26. L. 401: "2923"→"2023"

2.27. The "recovery period" is defined both 1997-2023 and 2000-2023. This should be consistent, also with a reference for the choice of year.

2.28. Supplement L. 36: "MIPAS_MA_-UA" → "MIPAS_MA_UA"